# Branching morphogenesis in the developing kidney is not impacted by nephron formation or integration

Kieran M Short[1], Alexander N Combes[2,3], Valerie Lisnyak[1], James G Lefevre[4], Lynelle K Jones[1], Melissa H Little[2,3,5], Nicholas A Hamilton[4], Ian M Smyth[1,6]*

[1]Department of Anatomy and Developmental Biology, Monash Biomedicine Discovery Institute, Monash University, Melbourne, Australia; [2]Murdoch Children's Research Institute, Parkville, Australia; [3]Department of Anatomy and Neuroscience, School of Biomedical Sciences, University of Melbourne, Parkville, Australia; [4]Division of Genomics of Development and Disease, Institute for Molecular Bioscience, The University of Queensland, Brisbane, Australia; [5]Department of Pediatrics, Faculty of Medicine, Dentistry and Health Sciences, University of Melbourne, Parkville, Australia; [6]Department of Biochemistry and Molecular Biology, Monash Biomedicine Discovery Institute, Monash University, Melbourne, Australia

**Abstract** Branching morphogenesis of the ureteric bud is integral to kidney development; establishing the collecting ducts of the adult organ and driving organ expansion via peripheral interactions with nephron progenitor cells. A recent study suggested that termination of tip branching within the developing kidney involved stochastic exhaustion in response to nephron formation, with such a termination event representing a unifying developmental process evident in many organs. To examine this possibility, we have profiled the impact of nephron formation and maturation on elaboration of the ureteric bud during mouse kidney development. We find a distinct absence of random branch termination events within the kidney or evidence that nephrogenesis impacts the branching program or cell proliferation in either tip or progenitor cell niches. Instead, organogenesis proceeds in a manner indifferent to the development of these structures. Hence, stochastic cessation of branching is not a unifying developmental feature in all branching organs.
DOI: https://doi.org/10.7554/eLife.38992.001

*For correspondence:
ian.smyth@monash.edu

## Introduction

Branching morphogenesis is an integral feature of the development of many organs including the kidneys, lungs and mammary gland. Branching typically begins with the formation of a bud like organ anlage which then proceeds to grow and divide, usually by bifurcation. The resulting network of interconnected epithelial tubules serves to break up the larger tissue masses in which they form, facilitating the exchange of oxygen (lungs) and nutrients (vasculature) and the transport of waste (kidneys) or secretions (glands). In some organs like the kidneys and lungs, the process of branching proceeds as a result of the orchestrated interaction of cells at the tips of the branching epithelium and specialized surrounding mesenchymal cells. In others, like the mammary gland, epithelial growth and arborisation occurs through collective cell migration in the absence of specific associated mesenchyma. In the kidneys, the ureteric bud (UB) elaborates (from embryonic day (E)11.5 in mice) within a field of specified cells known as the metanephric mesenchyme, which coalesces to form cap mesenchyme (CM) niches closely associated with each of the UB tips. These CM cells are nephron

**eLife digest** During development and before birth, many organs develop from branched tubes. Whether forming the airways of the lungs, the collecting ducts of the kidneys or the milk ducts of the breast, there are many similarities between these structures. Given their shared tree-like structures, one possibility is that these tissues all form through the same general process.

A key challenge is understanding why branched networks develop and pattern in such a way as to assume their functional roles in the adult organ. A unifying theory, which proposes that certain tips stop growing in a random manner, has been proposed to solve this problem. In this theory, the branched mammary gland structures stop growing when the tips of the structure impinge on neighbouring branches. In the kidney, this cessation has been proposed to occur when nephrons – the structures that filter urine from blood – form near the end of the collecting ducts.

By growing kidneys in the laboratory and studying developing kidneys in mice, Short et al. investigated whether nephrons do affect collecting duct growth and branch development. The results of these experiments instead suggest that nephron formation has no effect on duct growth or branching. The nephrons also do not appear to affect how quickly the duct cells grow and divide. Moreover, there is no evidence that the cell proliferation in individual branch tips ceases randomly by any other mechanism.

Overall, the experiments Short et al. performed suggest that a unifying theory of branching in developing organs may not hold true, at least not in the way that has been envisioned previously.
DOI: https://doi.org/10.7554/eLife.38992.002

progenitors (*Kobayashi et al., 2008*), sub-populations of which sequentially commit to form nephrons as development progresses. They do so by first forming pre-tubular aggregates which undergo mesenchymal to epithelial transition to form renal vesicles (RV) that differentiate into comma- and S-shaped bodies that elongate and mature to form nephrons. Because the network established by the branching UB will ultimately form the urine collecting system, the nascent nephrons reconnect to the tip from which they were generated at the comma stage and establish a patent lumen through which urine can be transported (*Kao et al., 2012*; *Georgas et al., 2009*).

The pervasive role of branching morphogenesis in shaping the development of many organs poses the question as to whether it is governed by shared mechanisms or features. A recent paper by Hannezo et al. has proposed a 'unifying theory of branching morphogenesis' (*Hannezo et al., 2017*). This work, based principally on studies in the mammary gland, posits that the elongating tips of the branching mammary tree randomly explore their environment but cease branching and proliferation when in proximity to neighboring branches. It further proposes that developmental morphogenesis in many organs occurs by employing similar stochastic, self-organized approaches characterized by branching and annihilating random walks (BARW). In the kidney, the cessation of branching explicit in the BARW model was proposed to derive from the differentiation of nephrons at individual UB tips. We have investigated a central tenet of this theory - the premise that nephron formation triggers cessation of branching. In doing so, we have examined the broader impact of nephron formation and integration on the behavior of both UB tip and nephron progenitor cell niches. The relationship between these populations is critically important for establishing nephron endowment in the adult kidney; a metric increasingly thought to influence susceptibility to kidney disease and broader physiological measures such as blood pressure (*Luyckx et al., 2013*). We find no evidence that nephron formation can influence the extent of ureteric branching in the developing kidney nor does it alter tip or cap cell proliferation. This indicates that stochastic cessation of branching does not operate during kidney development and demonstrates that nephrogenesis has little to no impact on the progress of branching morphogenesis or the behavior of tip or CM progenitor cells in the developing organ.

## Results and discussion

We have previously profiled the size and cell behavior of the CM and UB tip niches in the developing kidney and correlated this with the rate of branching in the organ (*Short et al., 2013*; *Short et al., 2014*). This study showed that branching is most rapid in the period between E12.5 and E17.5 but

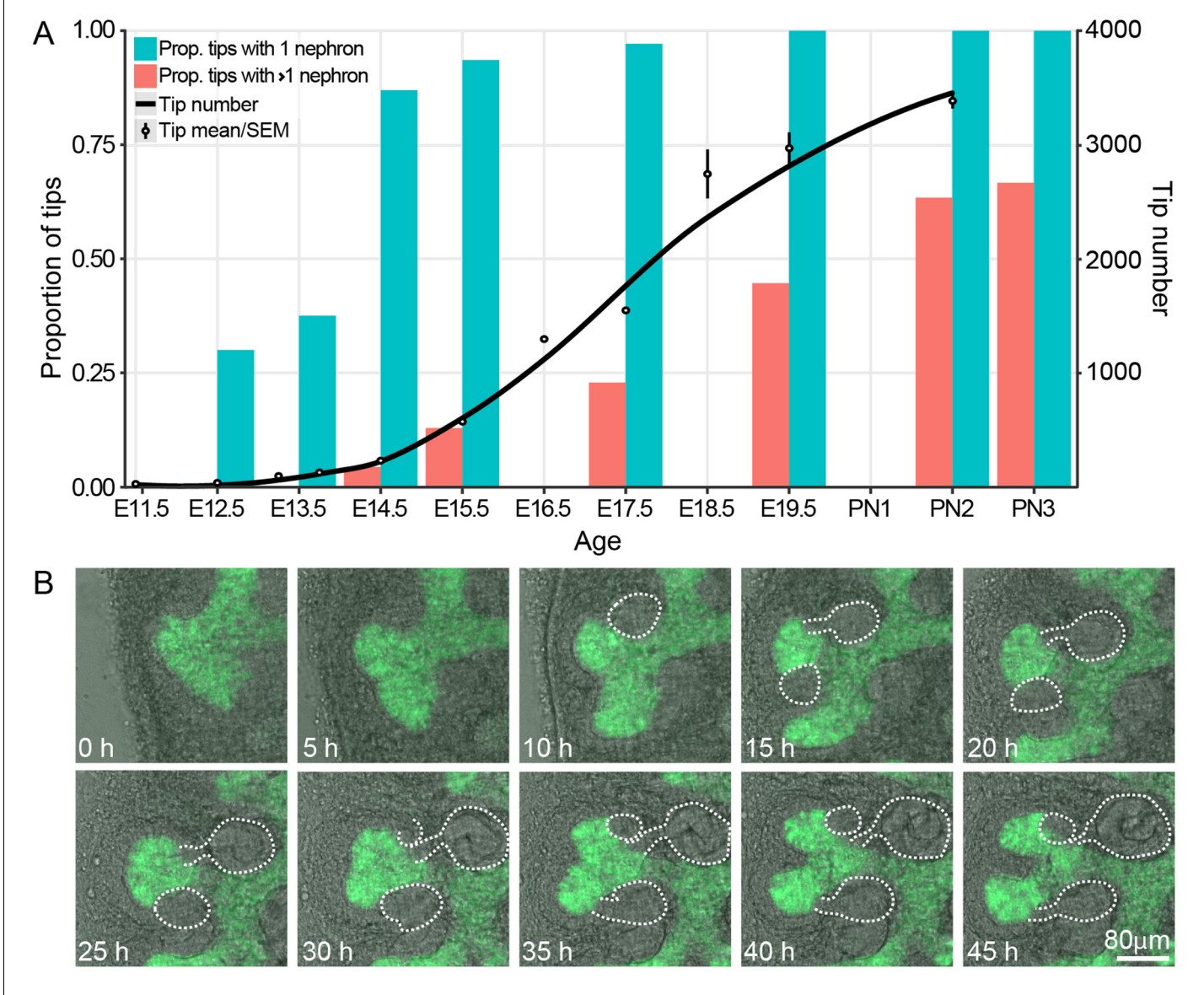

**Figure 1.** The proportion of ureteric bud tips associated with one or more connected nephrons as a product of developmental time. (**A**) Analysis of developing fetal and early postnatal kidneys in which the proportion of ureteric bud tips with 1 or more nephrons which had re-integrated into their associated ureteric bud tips are quantified (left y-axis, blue and orange bars; assessed at E12.5, 13.5, 14.5, 15.5, 17.5, 19.5, PN2 and PN3). The number of tips at each developmental stage is also shown (right y-axis; assessed at E11.5, 12.5, 13.25, 13.75, 15.5, 16.5, 17.5, 18.5, 19.5 and PN2). (**B**) Screen shots from a live imaging experiment of cultured Hoxb7-gfp kidneys tracking branching morphogenesis of the ureteric bud (green) and the formation and differentiation of renal vesicles into connected nephrons (white line). Time (h, hours) and scale are indicated.

DOI: https://doi.org/10.7554/eLife.38992.003

The following source data and figure supplement are available for figure 1:

**Source data 1.** Embryos were dissected at embryonic ages between E12.5 and E22.5 (the first column, embryonic day).
DOI: https://doi.org/10.7554/eLife.38992.005

**Source data 2.** Embryos were dissected at embryonic ages between E11.5 and E21.5 (the Embryonic Day column), and staged using limb staging (the Stage Column).
DOI: https://doi.org/10.7554/eLife.38992.006

**Figure supplement 1.** Screen shots from a live imaging experiment of cultured Hoxb7-gfp kidneys tracking branching morphogenesis of the ureteric bud (green) without overlays, showing nephron differentiation in phase contrast.
DOI: https://doi.org/10.7554/eLife.38992.004

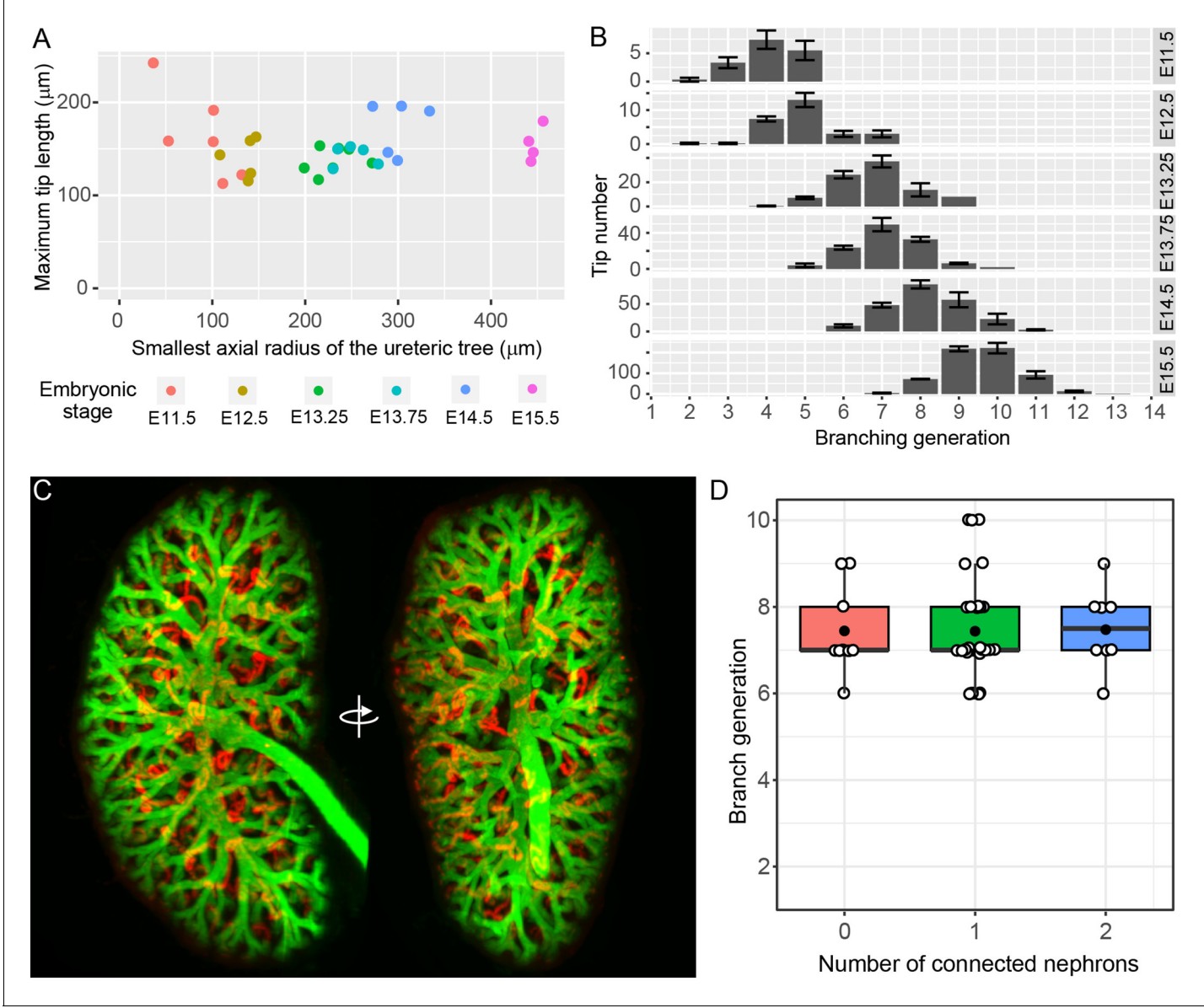

**Figure 2.** The relationship between branching morphogenesis and developmental time. (**A**) Analysis of maximal terminal tip length and axial radius of the kidney shows no relationship between the two measures (p=0.95, lm linear regression in R) at different developmental time points (where n = 6,5,7,5,5,4 for increasing time points, respectively). (**B**) The distribution of tip branching generations showing the mean and SEM of kidneys at each developmental stage (n = 6,5,7,5,5,4 for increasing time points, respectively). The branching generation of a tip is the number of branching events along the path from the ureter to the tip extremity determined using Tree Surveyor software. (**C**) A digital projection of light sheet imaging of a whole kidney at E15.5 labellling the collecting duct (green, Trop2[+]) and differentiating nephrons (red, E-cadherin[+];Trop2[-]). (**D**) The direct relationship between nephron number and branch generations per tip shows no evidence for a reduction in branching associated with nephron integration. Generation number of tips with no (red), 1 (green) and 2 (blue) connected nephrons are indicated. Each tip is represented (open circle); mean (closed circle) and median (bar) are indicated.

DOI: https://doi.org/10.7554/eLife.38992.007

The following source data is available for figure 2:

**Source data 1.** A single kidney from an E15.5 embryo was dissected, whole mount stained and analysed using light sheet microscopy and Imaris software, as described in the methods.
DOI: https://doi.org/10.7554/eLife.38992.008
**Source data 2.** Statistical analyses for *Figures 2C* and *3A*, and 3B.
DOI: https://doi.org/10.7554/eLife.38992.009

did not examine, in detail, the relationship between nephron formation and branching itself. To do so, we used high resolution confocal microscopy to profile the number of connected nephrons per tip from E12.5 through to the cessation of nephron formation at postnatal day 3 (PN3). One prediction of the annihilation model in which nephron formation triggers cessation of branching (*Hannezo et al., 2017*), is that nephron formation should not precede the branching process (which the model predicts nephron formation will halt). Analysis of our confocal data (*Figure 1A*) indicates that almost all the tips of the branching ureteric tree carry an attached nephron 72 hr after branching initiates (87% at E14.5) and that further nephrons are added as development (and branching) progresses. We next utilized organ explant cultures to provide supporting evidence that ongoing branching of tips with associated nephrons was indeed possible. Developing kidneys grown in culture lose the 3D structure of an in vivo organ, but retain local CM-UB interactions including branching morphogenesis and nephron induction (*Lindström et al., 2015*; *Combes et al., 2016*; *Watanabe and Costantini, 2004*). Using a Hoxb7-GFP transgene to visualise the ureteric epithelium and bright field illumination to identify forming nephrons across time, we observed tips with attached nephrons undergoing bifurcation (*Figure 1B*, *Figure 1— figure supplement 1*).

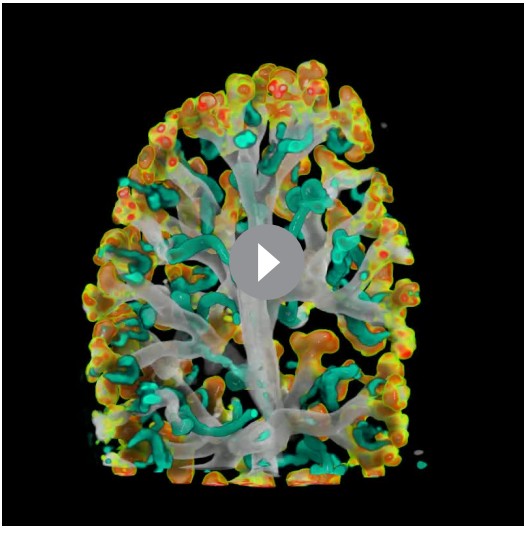

**Video 2.** Light sheet imaging of cell proliferation in UB tips of whole labelled kidney at E15.5 Whole mouse kidneys (E15.5) were labelled with antibodies to detect collecting duct and nephrons. Collecting duct volumes (greyscale) are registered alongside differentiating nephrons (cyan, generated by digital subtraction of Trop2 volumes from E-cadherin volumes). Density of EdU labelling within the ureteric bud volumes are indicated by a yellow-red heat map. Samples were imaged on an Ultramicroscope II (LaVision BioTec).
DOI: https://doi.org/10.7554/eLife.38992.011

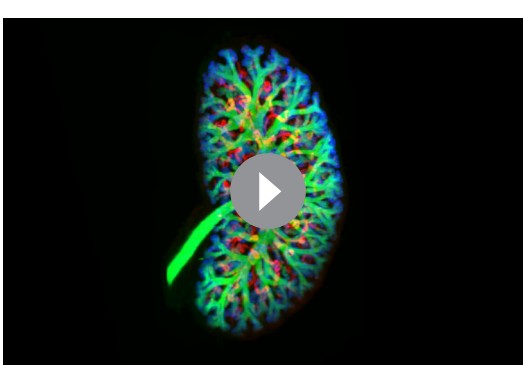

**Video 1.** Light sheet imaging of whole labelled kidney at E15.5 Whole mouse kidneys at E15.5 were labelled with antibodies to detect collecting duct and nephrons. Collecting duct volumes (green, Trop2 staining) are registered alongside differentiating nephrons (red, generated by digital subtraction of Trop2 from E-cadherin volumes) and incorporated EdU (blue, including only cells in the duct and nephrons volumes). Samples were imaged on an Ultramicroscope II (LaVision BioTec).
DOI: https://doi.org/10.7554/eLife.38992.010

All of the UB tips in the developing kidney are localized to the surface of the organ, where they remain throughout development (*Short et al., 2013*; *Short et al., 2014*; *Short et al., 2010*). Notwithstanding the results derived from organ culture, this observation raises the formal possibility that nephron formation might still impede the branching of tips - but not their growth. In this scenario, for any given tip in which branching (but not growth) had ceased, it would be necessary for the branch to lengthen over time in order to maintain its position on the organ surface. If extension of non-branching, nephron-associated tips were continuing we would expect that the maximum tip length would grow with the size of the kidney. Since only a small proportion of tips at any given stage might cease branching, we used the maximum tip length per organ to assess this possibility (rather than the mean tip length, which might mask any such effects). Using this approach, no evidence of maximum tip length extension was observed relative to the increase in axial radius of the fetal organ with age (*Figure 2A*, p=0.95). Furthermore, if branching were to cease in a subpopulation of developing

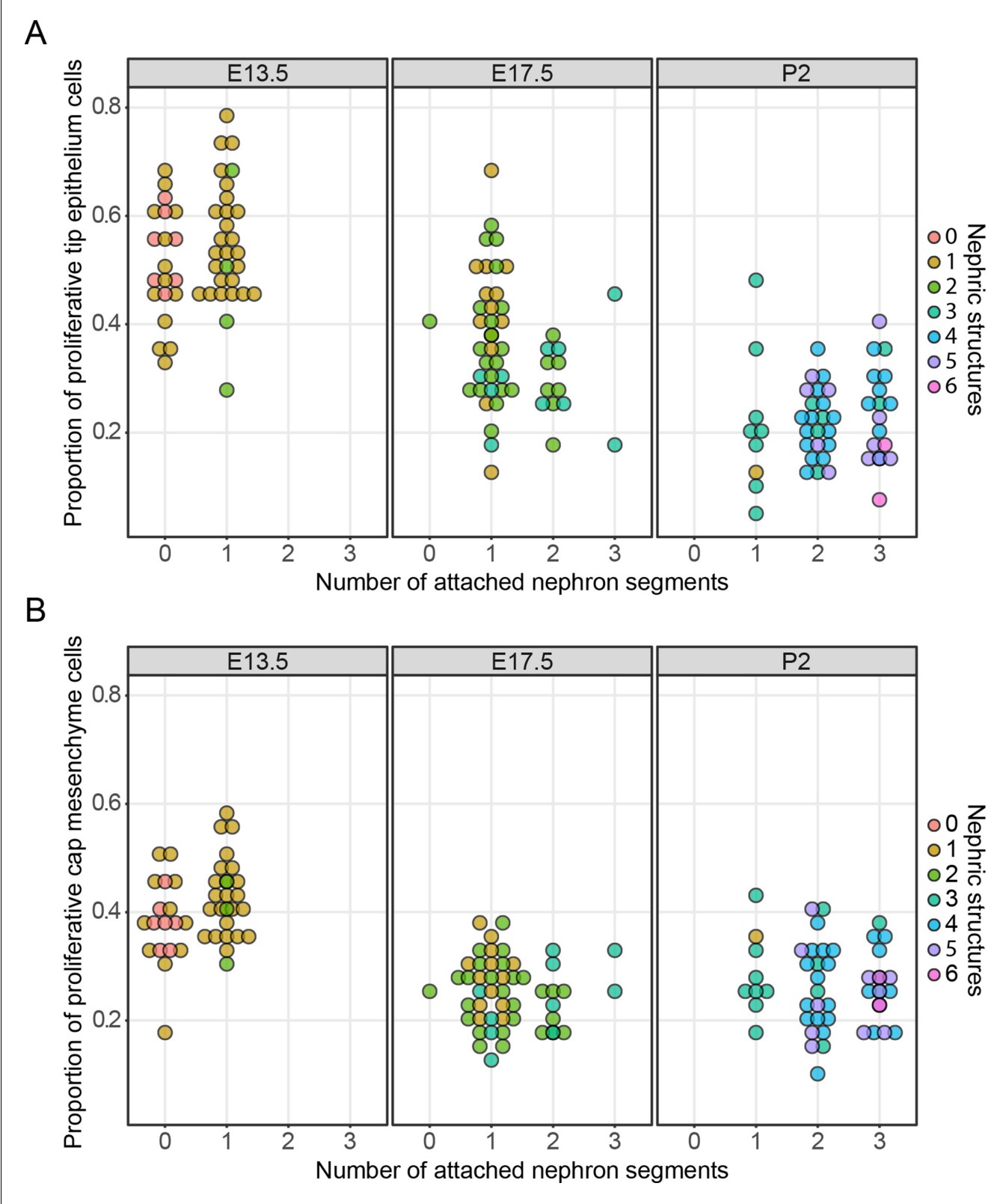

**Figure 3.** The relationship between cell proliferation nephron endowment. EdU labelling was used to quantify cell proliferation in the tip (A) and cap (B) cell niches. The number of nephrons, which are physically connected to the tips are detailed across three different developmental time points and the total number of nephric structures (connected and associated) are indicated in colour. Each data point represents a single cap or tip cell niche.

DOI: https://doi.org/10.7554/eLife.38992.012

*Figure 3 continued on next page*

*Figure 3 continued*

The following source data is available for figure 3:

**Source data 1.** Kidneys samples (column 1) were dissected from embryos at embryonic ages (column 2), and stained for Edu, Six2, Trop2 and DAPI.
DOI: https://doi.org/10.7554/eLife.38992.013

tips, their generation should remain static at subsequent stages. We therefore examined the distribution of tip branch generations (the number of branch segments between the root of the tree and each of the tips) but found no evidence of static branch generations. Instead, the distributions indicate continued branching of all tips (*Figure 2B*).

Simple measurements of nephron number per UB tip do not address how the presence of nephrons might have affected the historical branching behaviour of an *individual* tip. To better assess the relationship between the nephron 'phenotype' of a given tip and its branch history, we employed light sheet microscopy to image whole kidneys at E15.5 and then related the number of connected nephrons attached to a given tip with the number of bifurcations between that tip and the pelvis of the developing kidney (*Video 1*). Assessment of branch history of 50 tips in the kidney at this age found no evidence to support a relationship between the two measures (*Figure 2D*, Pearson's r = 0.014, p=0.9210). Moreover co-labelling for EdU labelled proliferating cells found that high levels of cell division are apparent in tips with connected nephrons (*Video 2*.)

Taken together, these results provide no support for a 'unifying theory' in which nephron formation is responsible for the stochastic cessation of ureteric branching. We then examined the broader effects that nephron formation had on cell behaviors associated with nephron formation. In the mammary gland, proximity-associated annihilation of branching has been associated with a profound decrease in cell proliferation (*Scheele et al., 2017*). To examine whether stochastic or more subtle differences in tip cell division might be driven by nephron formation, proliferating cells in embryos at E13.5 and E17.5 and pups at P2 were labelled with a pulse of the nucleotide analogue EdU and kidneys collected and stained for subsequent confocal imaging of nephrogenic niches. This analysis allowed us to directly relate cell proliferation with the number of connected and associated nephron structures. We found that cell division in the tip cell niches decreases as development progresses (*Figure 3A*; $p < 1e^{-5}$, one-way ANOVA between any two stages), which is consistent with our previously reported studies (*Short et al., 2014*). However, we found little association between cell division and the number of total or connected nephron structures (at E13.5, p=0.0587/0.79 (connected/ total nephron structures), n = 49; E17.5, p=0.155/0.0296, n = 50; P2, p=0.06687/0.4772, n = 50, one-way ANOVA). The only exception was at E17.5 where there was a weak but significant association between total associated structures, but this was not significant when examining connected nephrons. We then extended this analysis to profile the same measures in the nephron progenitors associated with each tip (*Figure 3B*) to determine whether the differentiation of nephrons might instead impact on the behavior of this cell niche. We found a similar (and previously described [*Short et al., 2014*]) time-dependent decrease in the rate of proliferation ($p < 1e^{-5}$, one way ANOVA), which is consistent with recent work detailing the progressive ageing of these cells and the accumulation of intrinsic characteristics, which affect their contribution to nephrogenic programs (*Chen et al., 2015*). However, we again found little relationship between the number of nephrons derived from and/or associated with a given cap cell niche and the rate of cell proliferation in that niche (E13.5, p=0.01827/0.3654 (connected/total nephron structures), n = 49; E17.5, p=0.3435/ 0.1503, n = 50; P2, p=0.06605/0.921, n = 50, one-way ANOVA). Together these analyses provide no compelling evidence for an association between nephrogenesis and tip/cap niche proliferation together at any one stage of development.

In this study, we have examined whether nephron formation may contribute to a process of stochastic cessation proposed as a unifying theory of branching morphogenesis. Multiple lines of evidence suggest that this is not the case. Integrated nephrons are a feature of almost all tips in the actively branching kidney and live imaging of cultured organs highlights active branching in the presence of attached nephrons. Moreover, hallmarks of ceased branching are not evident in the analysis of terminal tip lengths, the progressive distribution of branch generations over time, rates of cell division or in the correlation between nephron number and branching history of a given tip. In particular, we have found little evidence for a relationship between nephron number and cell proliferation

in either the tip cell niches (where the nephrons integrate) or in the nephron progenitor cells of the cap mesenchyme from which they derive. The latter observation suggests that nephron progenitors are fated to differentiate based on an internal 'clock like' mechanism which is consistent with recent studies (*Chen et al., 2015*). It remains to determine the manner in which individual nephrons remain associated with their tips of origin. Based on our findings, branching morphogenesis during kidney development is a process which differs considerably from the mammary gland. Although they develop in a superficially similar manner (i.e. through arborisation of an epithelium anlage) the cellular events which direct these processes and which shape branching are likely significantly different and not dictated by a 'unifying' developmental mechanism. Instead, we favour a model in which renal branching more closely mirrors that observed in the lung, another organ whose development is driven by reciprocal epithelial-mesenchymal interactions (*Morrisey and Hogan, 2010*).

# Materials and methods

**Key resources table**

| Reagent type (species) or resource | Designation | Source or reference | Identifiers | Additional information |
|---|---|---|---|---|
| Genetic reagent (Mus musculus) | Hoxb7-GFP | PMID: 10322632 | MGI:Tg(Hoxb7-EGFP)33Cos/J; RRID:IMSR_JAX:016251 | |
| Antibody | Pan-cytokeratin | Abcam | Abcam:AB11213; RRID:AB_297852 | (1:200) |
| Antibody | Ecadherin | Thermo Fisher | Thermo Fisher:13–1900; RRID:AB_86571 | (1:400) |
| Antibody | Trop2 | R and D Systems | R and D Systems:AF1122; RRID:AB_2205662 | (1:100) |
| Antibody | Six2 | Proteintech | Proteintech:115621AP; RRID:AB_2189084 | (1:600) |
| Commercial kit | Click-iT EdU Alexa Fluor 647 imaging kit | Life Technologies | Life Technologies:C10340 | |
| Software | Tree Surveyor | PMID: 23193168 | | |
| Software | Drishti | doi: 10.1117/12.935640 | | |

## Kidney samples

C57Bl6/J mouse embryos were collected and individually staged based on limb and developmental (Theiler) criteria as previously described (*Short et al., 2013*). Limb stages (*Wanek et al., 1989*) of 8, 9, 10, 11, 12 and 13 were treated as equivalent to E12.0, E12.5, E13.25, E13.75, E14.5, E15.5 and E16.5 respectively. E19.5 was counted as PN0. For analysis of nephron number per tip whole mount kidneys were stained with antibodies to the elaborating ureteric tree (pan-Cytokeratin or Ecadherin) and nephrogenic niches imaged by confocal microscopy following clearing in benyl alcohol:benzyl benzoate (*Combes et al., 2014*). Nephrons were classified from confocal volumes based on morphology and integration into the neighbouring tips. The number of tips profiled at each developmental stage were as follows: E12.5 (20), E13.5 (24), E14.5 (46), E15.5 (31), E17.5 (35), E19.5/PN0 (56), PN1 (41), PN2 (30). Total tip number from each developmental stage was counted using Optical Projection Tomography and Tree Surveyor software (*Short et al., 2013*) up E16.5 and by confocal imaging and counting of CM cell niches at later time points using previously described approaches (*Combes et al., 2014*). The number of kidneys profiled at each developmental stage to analyse tip number were as follows: E11.5 (6), E12.5 (5), E13.25 (7), E13.75 (5), E14.5 (8), E15.5 (7), E16.5 (4), E17.5 (3), E18.5 (2), E19.5/PN0 (3), PN2 (3).

## Kidney culture

Hoxb7-EGFP embryos (*Srinivas et al., 1999*) were isolated at E12.5 and cultured on the lower surface of a transwell membrane insert (Costar) in a glass-bottom dish (Mattek) in DMEM media (Sigma) containing 10% FCS (Sigma). Distance between the insert and coverslip was set at ~70 µm, controlled by a custom metal spacer between the rim of the dish, supplemented by 70 µm beads (Corpuscular Inc.), in a manner analogous to previously reported fixed-Z imaging approaches

(*Saarela et al., 2017*). Samples were cultured and imaged at 10x magnification for >40 hr on a Dragonfly spinning disc confocal (Andor) in a stage-top incubator at 37 degrees C with 5% $CO_2$. 80 micron Z-stacks composed of 1.95 µm steps were taken every half an hour for GFP and bright field channels. Forming nephrons were identified from bright field images as epithelial structures.

## Analysis of branching

Ureteric trees from a staged series of wild type mouse kidneys (n = 6, 5, 7, 5, 5, four for limb stages 7–12) were established using Tree Surveyor software (*Short et al., 2013*). Terminal tip length was measured from the final branch point along Tree Surveyor fitted B-splines to the tip extremity. The smallest axial radius was defined as half of the smallest dimension of the bounding box around the tip positions, where the bounding box is aligned according to the eigenvectors of the tip position covariance matrix. The branching generation of a tip is the number of branching events along the path from the ureter to the tip extremity established by the Tree Surveyor software.

## In vivo labelling of cell proliferation

Pregnant dams (for E13.5 and E17.5) and P2 pups were injected with EdU (50 mg/kg) and kidneys collected following a 30 min chase period (*Combes et al., 2014*). Organs were then stained with antibodies to Six2 (Proteintech 115621AP), Trop2 (R and D systems AF1122)/pan-Cytokerin (Abcam AB11213) and EdU was detected using a Click-iT EdU Alexa Fluor 647 imaging kit (Life Technologies). Samples were cleared in benzyl alcohol:benzyl benzoate (Sigma Aldrich) at a ratio of 1:2 were then imaged using a Leica SP8 confocal microscope at 20X with z-stacks using ~1 AIRY unit spacing per wavelength, and tip and cap niches defined and characterized using reported protocols (*Combes et al., 2014*). Briefly, total and proliferating cell number was quantified in tip and cap niches manually masked using Imaris software (Bitplane).

## Light sheet imaging

Dissected fetal kidneys at E15.5 were stained with DAPI and antibodies to Six2 (Proteintech 115621AP), Trop2 (R and D systems AF1122), E-cadherin (Thermo Fisher 13–1900) and EdU was detected as previously described (*Combes et al., 2014*). Samples were then imaged on an Ultramicroscope II (LaVision BioTec) at ~4 x using a 0.52 µm z-step. 50 tips were then analysed for the presence and number of integrated nephrons (identified morphologically) and the branch history of each tip traced back to the renal pelvis. The relationship between tip nephron number and branch depth was assessed using R with a Pearson's product-moment correlation. In order to generate images and videos of whole mount branching with nephrons, the Trop2 signal data (strongly staining ureteric tree only) were subtracted from the E-cadherin signal data (strongly staining both ureteric Tree and mature nephrons) using the Image J image calculator. These data were visualized using Imaris and Drishti (*Limaye, 2012*).

# Acknowledgements

This work was supported by the National Health and Medical Research Council of Australia (NHMRC, APP1002748, APP1063696), the Human Frontiers in Science Program (RGP0039/2011) and the Australian Research Council (DP160103100). Microscopy was performed at Monash University (Monash Micro Imaging Platform) and the Murdoch Children's Research Institute. ANC was supported by a Discovery Early Career Researcher Award from the Australian Research Council. MHL is a Senior Principal Research Fellow and IMS is a Senior Research Fellow of the NHMRC.

# Additional information

### Competing interests

Melissa H Little: has consulted for and received research funding from Organovo Inc. The other authors declare that no competing interests exist.

## Funding

| Funder | Grant reference number | Author |
|---|---|---|
| National Health and Medical Research Council | 1002748 | Melissa H. Little |
| Australian Research Council | DP160103100 | Nicholas A. Hamilton<br>Ian M. Smyth |
| Human Frontier Science Program | RGP0039/2011 | Melissa H. Little<br>Ian M. Smyth |
| National Health and Medical Research Council | 1063696 | Melissa H. Little |

The funders had no role in study design, data collection and interpretation, or the decision to submit the work for publication.

### Author contributions

Kieran M Short, Conceptualization, Data curation, Formal analysis, Investigation, Methodology, Writing—original draft, Project administration, Writing—review and editing; Alexander N Combes, Conceptualization, Data curation, Formal analysis, Investigation, Methodology, Writing—original draft, Writing—review and editing; Valerie Lisnyak, Data curation, Formal analysis, Investigation; James G Lefevre, Conceptualization, Data curation, Formal analysis, Investigation, Methodology, Writing—review and editing; Lynelle K Jones, Data curation, Investigation; Melissa H Little, Conceptualization, Data curation, Supervision, Funding acquisition, Writing—review and editing; Nicholas A Hamilton, Conceptualization, Formal analysis, Supervision, Funding acquisition, Investigation, Methodology, Project administration, Writing—review and editing; Ian M Smyth, Conceptualization, Data curation, Formal analysis, Supervision, Funding acquisition, Investigation, Methodology, Writing—original draft, Project administration, Writing—review and editing

### Author ORCIDs

Ian M Smyth (iD) https://orcid.org/0000-0002-1727-7829

### Ethics

Animal experimentation: All animal experiments in this study were assessed and approved by Monash University or the Murdoch Children's Research Institute Animal Ethics Committees (MARP/2016/144) and were conducted under applicable Australian laws governing the care and use of animals for scientific purposes.

### Decision letter and Author response

Decision letter https://doi.org/10.7554/eLife.38992.016
Author response https://doi.org/10.7554/eLife.38992.017

## Additional files

### Supplementary files

• Transparent reporting form
DOI: https://doi.org/10.7554/eLife.38992.014

### Data availability

All data generated or analysed during this study are included in the manuscript and supporting files.

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
