## [Decision Letter]

Thank you for submitting your article "Branching morphogenesis in the developing kidney is not impacted by nephron formation or integration" for consideration by *eLife*. Your article has reviewed by Didier Stainier as the Senior Editor, a Reviewing Editor, and three reviewers. The following individual involved in review of your submission has agreed to reveal his identity: David Bryant (Reviewer #3).

The reviewers have discussed the reviews with one another and the Reviewing Editor has drafted this decision to help you prepare a revised submission.

Summary:

This manuscript seeks to test a "unifying theory of branching morphogenesis" proposed by Hannezo et al., 2017, which built from experimental and computational work in the mammary gland to argue that the cessation of branching in the developing kidney was triggered by the differentiation of nephrons at UB tips. The authors use a combination of powerful imaging methods, on cultured and fixed kidneys, to examine the relationship between ureteric bud tip branching and the formation of nephrons and their connection to the ureteric bud tips. The data show convincingly and quantitatively that ureteric bud growth, ureteric bud branching, and the proliferation of ureteric bud tip cells are all unaffected by nephron formation and connection. The reviewers were unanimous in finding that the study and manuscript demonstrate that the hypothesis from the Hannezo paper fails to explain branching morphogenesis in the kidney. Given the previous claims and the desire to identify general principles to explain branching across organs, this manuscript is timely and important and appropriate for the broad audience of *eLife*. In conclusion, the authors convincingly demonstrate – at least for the developing mouse kidney – that ureteric bud branching is not terminated by the formation of the nephron. In the physical sciences, it is common for theoretical work to motivate new experiments to test the limits of applicability of the theory. It is gratifying to have such an exchange in the field of organogenesis.

None of the reviewers identified essential experiments necessary for acceptance but the following minor points should be addressed.

1) Results and Discussion section, third paragraph. Figure 2D should be cited here, not Figure 2C.

2) Results and Discussion section. "We extended this analysis to […] nephron progenitor cells". It is not obvious to me how proliferation of nephron progenitor cells is directly relevant to the model being tested, so this experiment requires a better justification in the text.

3) I think the ordinates of the graphs in Figure 3 should be labeled "Proportion of proliferative tip epithelial cells" and "Proportion of proliferative cap mesenchyme cells".

4) Subsection “Kidney culture”. Please indicate the source of the 70 μm beads.

5) There was confusion amongst the reviewers as to whether data was collected for E16.5, E18.5 and PN1. If so, it should be supplied. If not, these labels should be removed from the X axis or it should be indicated somewhere that they were not examined. The reviewers found the trend in the biology clear from the other time points and so if the data are not available, new experiments are not essential.

6) Regarding Figure 1B, it was difficult for the reviewers to see the condensing mesenchyme into renal vesicles from the transmitted light images. Perhaps these could be presented in a clearer fashion, as they are likely evident in the bright field images. One problem might be that the white dotted lines can actually make it harder to see the nephrons, so they might also include the images without the white lines (perhaps bright field images without white lines and without the GFP (either below each image with white lines and GFP, or else in a supplementary figure).

---

## [Author Response]

Essential revisions:1) Results and Discussion section, third paragraph. Figure 2D should be cited here, not Figure 2C.

Our apologies, this has been corrected in the revised manuscript

2) Results and Discussion section. "We extended this analysis to […] nephron progenitor cells". It is not obvious to me how proliferation of nephron progenitor cells is directly relevant to the model being tested, so this experiment requires a better justification in the text.

This assessment was really to see whether the differentiation of nephrons impacted on progenitor cell populations (as clearly it has little effect on the branching UB). To clarify this we have included an addition to this sentence.

It now reads:

“We then extended this analysis to profile the same measures in the nephron progenitors associated with each tip (Figure 3B) to determine whether the differentiation of nephrons might instead impact on the behavior of this cell niche.”

3) I think the ordinates of the graphs in Figure 3 should be labeled "Proportion of proliferative tip epithelial cells" and "Proportion of proliferative cap mesenchyme cells".

We concur – this has been changed.

4) Subsection “Kidney culture”. Please indicate the source of the 70 μm beads.

This information has now been added.

5) There was confusion amongst the reviewers as to whether data was collected for E16.5, E18.5 and PN1. If so, it should be supplied. If not, these labels should be removed from the X axis or it should be indicated somewhere that they were not examined. The reviewers found the trend in the biology clear from the other time points and so if the data are not available, new experiments are not essential.

We presume this comment refers to Figure 1A. Tip numbers were counted at the time points indicated by circles (with reference to the above comment, at E16.5 and E18.5) whilst the nephrons were counted at the time points indicated by the coloured graphs. No data was collected at PN1, but we wanted to maintain an equal scale in terms of developmental time on the X-axis so we included it. We have changed the Figure legend to detail exactly when each of these measures was assessed and hope that this serves to clarify things.

It now reads:

“(A) Analysis of developing fetal and early postnatal kidneys in which the proportion of ureteric bud tips with 1 or more nephrons which had re-integrated into their associated ureteric bud tips are quantified (left y-axis, blue and orange bars; assessed at E12.5, 13.5, 14.5, 15.5, 17.5, 19.5, PN2 and PN3). The number of tips at each developmental stage is also shown (right y-axis; assessed at E11.5, 12.5, 13.25, 13.75, 15.5, 16.5, 17.5, 18.5, 19.5 and PN2).”

6) Regarding Figure 1B, it was difficult for the reviewers to see the condensing mesenchyme into renal vesicles from the transmitted light images. Perhaps these could be presented in a clearer fashion, as they are likely evident in the bright field images. One problem might be that the white dotted lines can actually make it harder to see the nephrons, so they might also include the images without the white lines (perhaps bright field images without white lines and without the GFP (either below each image with white lines and GFP, or else in a supplementary figure).

We concur, though it is a little tricky to display given that the structures are in different planes. We have now included Figure 1—figure supplement 1 without the overlay which we think better shows the formation and positioning of these structures. If the reviewers disagree we could look at other options (stripping out the gfp, presenting other views in z etc.). We are happy to be guided by the reviewers in this respect.